# Protein Binding: A Fuzzy Concept

**DOI:** 10.3390/life13040855

**Published:** 2023-03-23

**Authors:** Mike P. Williamson

**Affiliations:** School of Biosciences, University of Sheffield, Firth Court, Sheffield S10 2TN, UK; m.williamson@sheffield.ac.uk

**Keywords:** affinity, intrinsically disordered protein, avidity, effective concentration, facilitated dissociation, allovalency, fuzzy binding, protein-protein interaction

## Abstract

Our understanding of protein binding interactions has matured significantly over the last few years, largely as a result of trying to make sense of the binding interactions of intrinsically disordered proteins. Here, we bring together some disparate ideas that have largely developed independently, and show that they can be linked into a coherent picture that provides insight into quantitative aspects of protein interactions, in particular that transient protein interactions are often optimised for speed, rather than tight binding.

## 1. Introduction

In 1894, Emil Fischer proposed the lock-and-key model in order to make sense of the high degree of specificity in carbohydrate biochemistry [1]. He proposed that proteins had specific shapes that fitted exactly to the shapes of their substrates, and that this lock-and-key matching was the explanation for their remarkable substrate specificity. This idea is still very useful, but has been refined since. Our improving understanding of protein structure and dynamics led first to the induced fit model, and then to the conformational selection model. The consensus view now is that both proteins and their ligands change conformation as the binding develops, so that binding has features of both models. This implies that binding is not a simple matching of key into lock, but is more of a mutual adapting of the two molecules, which in detail is neither induced fit nor conformational selection. The concept has been described appropriately as conformational funnelling, by analogy with the energy landscape concept of the protein folding funnel in which proteins gradually gain conformational order as they fold [2]. Over the last 20 years, it has become apparent that many proteins (often stated to be as high as one third of the mammalian genome) contain significant regions of intrinsically unstructured polypeptide in their native state. This has forced us to think more carefully about how such proteins can have specific functions. There is somewhat less consensus here, but the general view is that these intrinsically disordered proteins (IDPs) or intrinsically disordered regions (IDRs) have a wide range of disorder, with some being genuinely disordered and others having significant populations of partially folded states (for example, helical regions), and similarly that in the bound state, some IDPs become fully ordered, while others retain a large amount of disorder, but that probably all IDPs become more ordered on binding [3]. This clearly widens the types of events that we should consider functionally important binding.

In the following, we propose a unifying concept that for transient protein binding interactions, the most important factor is often not affinity (tightness of binding), but rates (on-rate and more particularly off-rate). We then apply this idea to increasingly more complex types of interaction and show how it can produce a wide range of effects, some of which are counter-intuitive. This paper focuses on transient complexes, by which we mean complexes that bind and dissociate frequently as an intrinsic part of their function [4]. Specifically, we do not consider the interactions between proteins that form obligate oligomers and seldom dissociate.

## 2. Results

### 2.1. Proteins Are Adapted for Fast Interactions, Not Strong Ones

There is a clear link between rates and affinities. For a simple bimolecular interaction,
*k*_on_
P + L ⇌ PL
*k*_off_
where P is protein, L is ligand and [P] and [L] are their concentrations, the forward rate of binding is [P][L]*k*_on_ and the backward rate of dissociation is [PL]*k*_off_. At equilibrium, the forward and backward rates are equal, which means that
[P][L]*k*_on_ = [PL]*k*_off_
(1)
or equivalently
(2)$$\frac{k_{\mathrm{off}}}{k_{\mathrm{on}}}=\frac{\left[P\right]\left[L\right]}{\left[PL\right]}=K_{d}$$
where *K_d_* is the dissociation constant and has units of concentration, and is equal to 1/*K_a_*, the association constant (with units of concentration^−1^). Specifically, this shows that binding affinities and on- and off-rates are interdependent: one cannot change one without changing the others. There is a physical limitation to the on-rate, often described as the diffusion limit: the two molecules have to meet in order to bind, which is ultimately limited by their diffusion rates. For two small hard spheres in water, the diffusion limit is roughly 10^9^ M^−1^s^−1^. For a folded protein binding to a ligand, the rate will inevitably be considerably less than this, firstly because the ligand has to collide at the binding site (meaning that the rate is reduced by the fraction of the protein surface area that comprises the binding site), and secondly because there almost always has to be some rearrangement of the protein in order to bind, as discussed above. For two proteins binding, the rate is also reduced because diffusion rates slow down as molecular size increases. Thus, in practice, the fastest possible on-rate for a globular protein is more like 10^6^–10^7^ M^−1^s^−1^. This implies that the fastest possible off-rate for a globular protein is approximately *K_d_* × 10^7^ s^−1^.

A typical “weak” protein interaction might have a *K_d_* in the μM range, while a “strong” interaction could be 1000 times stronger or in the nM region. If *K_d_* is 1 nM and *k*_on_ is 10^6^ M^−1^s^−1^, then the off-rate is *k*_on_.*K_d_* = 10^−3^ s^−1^, or in other words, the ligand takes of the order of 1000 s or 20 min to dissociate. For most biological interactions, this is far too slow. In order for dissociation to occur at a rate compatible with the biology, there are several options: the on-rate could be sped up, ligand dissociation could be encouraged to go faster in some way, or the overall affinity could be weakened. We will see examples of all of these below, but this is probably the main reason that most proteins do not bind tightly to their ligands. There is no inherent physical limitation to very tight protein–ligand interactions if required: witness, for example, the streptavidin–biotin affinity of about 10^−14^ M, or bacterial colicin–immunity protein complexes, which have affinities of around 10^−15^ M [5]. In other words, proteins could bind very tightly to their ligands if this were required, and the fact that they do not bind tightly indicates that some other factor than affinity is dictating the features of the interaction. We suggest that this is kinetics, and specifically the need for a fast off-rate.

It is worth noting that this limitation on tight affinities has a major implication for the concentration of hormones in the blood. In order to turn on its receptor, the concentration of a hormone has to change from about 10× less than its *K_d_* to 10× more than its *K_d_* [6]. This limitation on off-rates dictates that hormone concentrations cannot normally be lower than about 10 pM. As a specific example, we note that the affinity of epinephrine and norepinephrine (adrenaline and noradrenaline) for their receptors is typically around 300 nM, implying concentrations 10× larger than this (i.e., 3 μM) to turn on the receptors to a level of 90%. Even this rather high concentration translates to off-rates of the order of 0.05 min^−1^, which seems an uncomfortably slow rate considering the widespread physiological effects of epinephrine [7].

A further indication of the importance of a fast off-rate comes from consideration of enzyme turnover rates. The uncatalysed rates of biological reactions cover an enormous range, from slower than 10^−20^ s^−1^ (phosphoryl transfer [8]) to about 0.1 s^−1^ (carbonic anhydrase). However, enzyme-catalysed turnover rates are all around 10^3–^10^5^ s^−1^ [9]. In the rather limited number of cases where the enzyme catalysis has been studied in sufficient detail, it is clear that the rate-determining factor for the turnover rate is the dissociation rate of products. In other words, the critical factor determining how fast enzyme reactions go is not the reaction itself, but the rate at which ligands dissociate. The main reason for enzymes to equilibrate between open and closed conformations (“induced fit”) is probably to allow substrates to bind and products to leave at a reasonable rate, rather than the formation of a suitable conformation for catalysis.

### 2.2. Intrinsically Disordered Proteins Have Fast On-Rates

The binding of IDPs to their targets has several interesting features. The affinity is generally weak (typically around 0.1 μM, making it in general slightly weaker than typically affinities between globular proteins) [10]. The surface area covered by the IDP on its receptor is usually small and relatively flat. This means that on-rates can be very fast: a recent review identified on-rates of 10^9^–10^10^ s^−1^, which are at or even faster than the diffusion limit [11]. It was suggested that this very fast on-rate may arise from a combination of factors, not all of which may apply in each case. The bound state of the IDP often retains considerable flexibility: transient formation of secondary structures in the free state may speed up binding, and in some cases there are clear rate increases that arise from electrostatic interactions. “Guiding” electrostatic interactions have been observed in several protein–protein interactions, which may operate partly from simple electrostatic attraction, but are also likely to use dipole orientation to orient the proteins before binding [12,13]. However, in IDPs, a novel feature has been identified: electrostatic interactions may reduce the dissociation rate [14]. In this context, it is worth noting that the binding of many IDPs is regulated by post-translational modifications, in particular by multiple phosphorylation [15], which could work in this way (see below). In addition, there has been a proposal that IDPs may have a faster on-rate due to a phenomenon named fly-casting, by which the IDP can reach out to a binding partner, bind transiently, and “reel it in” [16]. This idea is attractive, but remains to be proven with any confidence.

It seems self-evident that a protein that has to fold (at least partially) on binding must therefore bind more weakly than a protein that is already folded before it binds. The evidence suggests that this is indeed true: one survey found that binding affinities were weaker for IDPs than for folded proteins by approximately 10 kJ/mol, which the authors noted would be the energy required to go from 2% folded to 100% folded [17]. The fact that so many proteins do this suggests that it has a functional benefit: an obvious benefit is that it enables them to bind and dissociate more rapidly, and that this is more important than ensuring strong binding.

Having introduced some unifying concepts, we now turn to consider a range of increasingly complex interactions.

### 2.3. Even for Simple Bimolecular Interactions, the Bound State Is Dynamic

Binding affinity is the ratio of concentrations of free and bound species, or equivalently the ratio of off- and on-rates (Equation (2)). When a protein binds its ligand, there is invariably some mutual conformational and energetic adjustment. This means that it is not entirely clear at what stage a free protein becomes bound or a bound protein becomes free. It is helpful to consider two stages in the binding process. The first is the encounter complex, which is the first stage at which protein and ligand come into contact, prior to any desolvation. Simulations (discussed in [18]) show that encounter complexes can last for several nanoseconds, long enough for both partners to reorient significantly and explore a range of bound geometries before dissociation. The lifetime of the encounter complex depends significantly on the charges on both partners. The second stage is the transition state for binding, which is defined in the classical manner as the state from which half the complexes dissociate and half go on to bind constructively. For interactions between two globular proteins, the encounter complex usually comes first (though clearly there will be interactions within the ensemble that proceed rapidly on to binding, and for which the transition state comes first). For IDPs, which more typically bind at or close to the diffusion-controlled limit, the transition state comes first, because diffusion-controlled on-rates imply that almost every collision leads successfully to binding. The stages between the transition state and the bound complex for IDPs must therefore necessarily involve a greater or lesser amount of coupled folding and binding [3].

For techniques such as surface plasmon resonance or microscale thermophoresis, it would be reasonable to assume that binding has occurred once protein and ligand have committed to being attached, or in other words once the encounter complex has formed [18,19]. For other techniques, such as isothermal calorimetry or UV/fluorescence spectroscopy, the change measured is only achieved once the fully bound conformation has been reached, which could be much later. It is therefore likely that different techniques will record different affinities, and it is remarkable that most techniques do generate roughly similar values. We note that there is some variation in the end-point, even when using essentially only a single technique [20].

NMR is an interesting technique for the measurement of affinity. The most common way of measuring affinities by NMR is to carry out a titration of ligand into protein, following binding by measuring chemical shift changes in ^15^N HSQC spectra [21]. This means that in principle, every H or N nucleus could generate a different affinity. We recently showed that there is indeed a range of affinities at different sites, varying by approximately a factor of two [22]. This variation in affinity is most apparent for proteins that have a very restrictive and rigid binding site, in which case the tighter affinities are seen for residues within the binding site, while residues around the edge (where ligand flexibility can be accommodated) have weaker affinities, because they see a more dynamic, and hence a more weakly bound, complex (Figure 1). We suggested that the affinity observed by other techniques approximates to the weaker affinity, corresponding to the average affinity of the whole ligand. In other words, even for a reasonably well-defined bimolecular complex, there is not a single binding affinity, but a spread of affinities at different sites on the protein, as a result of the flexibility of both protein and ligand. The same idea can be expressed by saying that because there is some disorder in the bound state, the binding affinity varies depending on the degree of disorder at each location on the protein surface.

### 2.4. Two Binding Sites Generate Stronger Affinity Because They Increase the Effective Ligand Concentration

The classic example of this effect is antibodies: IgG antibodies have two Fab regions, each of which can bind to an antigen. In that context, the effect is often described as avidity (Figure 2). Once one Fab is bound, then the other is restrained to be close to it. If the two antigens are also kept close together in space (by being in the same molecule or attached to the same membrane surface), then the effect is to increase the overall affinity. This is essentially a contrast between intermolecular binding of Fab to antigen and intramolecular binding, when Fab and antigen are already attached to each other at one end. This is often expressed in terms of the effective concentration *C*_eff_ of the second Fab for its antigen. The effect is often dramatic. Bobrovnik [24] provides a simple example: a receptor present at a concentration of 10 nM, that has two binding sites for two ligands, one with an affinity of 3 μM and one with an affinity of 1.2 μM. If the two ligands are unconnected, then only about 0.002% of all receptors bind both ligands. However, if the two ligands are connected by a linker of suitable length to match the separation of binding sites on the receptor, then about 98% of receptors now bind both. Avidity is thus highly effective.

How can *C*_eff_ be calculated? Because the on-rate is the rate constant multiplied by the concentration, and it is reasonable to assume that intermolecular binding and intramolecular binding have the same rate constant, then
(3)$$\frac{on-rate_{intra}}{on-rate_{inter}}=\frac{C_{\mathrm{eff}}k_{\mathrm{on}}}{\left[Fab\right]k_{\mathrm{on}}}=\frac{C_{eff}}{\left[Fab\right]}$$

If we further make the not-unreasonable assumption that the off-rates for the intermolecular and intramolecular binding are identical, then the on-rates are inversely proportional to the affinities, or
(4)$$\frac{K_{inter}}{K_{intra}}=\frac{\left[Fab\right]}{C_{\mathrm{eff}}}$$

In other words, the affinity for binding of the second domain when the first is already bound is strengthened by a factor of [Fab]/*C*_eff_. This shows that *C*_eff_ measures the factor by which the affinity is strengthened as a result of the avidity effect. It is thus a very useful number. An equivalent statement [24,25] is to say that if two unconnected ligands bind with affinities *K*_d1_ and *K*_d2_, but the connected ligands bind with a stronger affinity *K*_d_, then
*C*_eff_ = *K*_d1_    *K*_d2_/*K*_d_
(5)

This equation is helpful, because it shows that avidity starts to become an important consideration when *C*_eff_ reaches similar magnitudes to typical values of the weaker of the two binding affinities, i.e., low mM or stronger. It is therefore useful to be able to estimate it.

For many interactions, it is a difficult number to measure experimentally. There have been various methods proposed to calculate it, and more recently there have been some useful experimental measurements, which come together to derive a coherent understanding.

A conceptually simple way to calculate *C*_eff_ is to estimate the maximum distance between the two binding sites on the flexible ligand and use this to define a sphere (Figure 2). This is the volume that must contain the second ligand site, and so *C*_eff_ is simply equivalent to one molecule within that volume, or 1/(N_Av_ × 4/3 π*r*^3^), where N_Av_ is Avogadro’s number (6 × 10^23^) and *r* is the maximum distance between ligand sites [24]. This is likely to be a large underestimate of the true value of *C*_eff_, because peptide linkers are fairly rigid, and thus significant regions of the possible sphere are unlikely to be accessible. In particular, the calculation above implies that *C*_eff_ is strongly dependent on the length of the linker, being proportional to *r*^−3^. A more realistic model of peptide flexibility suggests that this sensitivity to peptide length is too extreme. Krishnamurthy et al. [26] noted that the expectation length of a random polymer chain is proportional to the square root of the number of monomers, which would imply something closer to *r*^−2^. However, the best way to determine the actual sensitivity is to measure it.

Krishnamurthy et al. [26] measured the dependence of *C*_eff_ on linker length for an ethylene glycol linker, and showed it to be close to 1/*n*, in that an increase in the number of monomers in the chain from 2 to 20 reduced *C*_eff_ by a factor of only 8. More recently, similar measurements have been made for the more relevant peptide linkers, demonstrating a relationship close to *n*^−1.5^ [27]. The sequence of the linker not surprisingly affects *C*_eff_ strongly: charged residues (and to some extent prolines) in the linkers make the linkers more extended, and therefore increase the sensitivity to *n*. However, in a more realistic model [24], the effect was much closer to *n*^−1^. This means that the linker length has a relatively small effect on *C*_eff_ and thus on avidity. A similar conclusion was reached by [28], who showed that increasing the linker length from 40 residues to 60 residues reduced *C*_eff_ by a factor of only 3 (from 0.6 mM to 0.2 mM), or *n*^−1.6^. Thus, there is clearly some variability depending on the peptide sequence and conditions [28], but the length of the linker makes surprisingly little difference. If the linker is so short that the ligand cannot reach both binding sites at the same time, then the affinity becomes weaker [29], ultimately losing any avidity [28]. However, a change in length of a linker that is longer than the minimum necessary has relatively little effect on overall affinity [26].

### 2.5. Facilitated Dissociation Enables a Faster Off-Rate

The cooperativity produced by avidity discussed above highlights the contrast between a protein that binds its ligand at a single site and one that can achieve the same overall affinity by binding at two sites, each of which has a much weaker affinity in isolation. The latter is a common observation, implying that it must bring advantages. Several have been proposed, as follows.

The weaker binding interactions inherently require less stringent geometrical matches between receptor and ligand. This permits a faster on-rate, because the geometrical requirements that limit the diffusion-controlled on-rate are less severe. If the overall affinities are maintained, this implies that two-site binding produces both a faster on-rate and a faster off-rate. As we argue throughout, this is a key desirable for any binding interaction.Although the overall off-rate for complete dissociation is still slow, because it depends on the overall affinity, the off-rate at each binding site is much faster. This is of particular benefit for enzymes that operate on macromolecular substrates. For example, many polysaccharide-digesting enzymes consist of a catalytic domain plus two or more carbohydrate binding domains. Each of these binds weakly, and thus detaches rapidly (Figure 3). Because one binding domain usually remains attached when the other detaches, this does not lead to complete dissociation of the enzyme from its substrate, which would be undesirable. However, it does allow the dissociated domain to rebind at a different location, and thus allow the enzyme to move around across the surface of its macromolecular substrate at a rate tuneable by the individual affinities and the lengths of the connecting linkers.The mobility shown in Figure 3, where the two binding sites are able to detach transiently, permits a marked acceleration of the overall rate of detachment by a mechanism that has been described as facilitated dissociation. This has been described by a number of authors [30,31,32], but the most well-characterised example is from the Wright–Dyson lab [33,34], which we now discuss.

The transcription factor HIF-1α controls the cell’s response to hypoxia. To do so, it binds to a TAZ-1 domain, common to several transcriptional coactivators. This binding competes with the binding of TAZ-1 to CITED2. The structure of TAZ-1 is a fairly well-defined helical bundle, but both HIF-1α and CITED contain significant disordered regions. The binding affinities of TAZ-1 to both HIF-1α and CITED2, when measured individually, are similar—about 10 nM. However, remarkably, if TAZ-1 is mixed with equal concentrations of HIF-1α and CITED2, the resultant complex contains 100% CITED2. In other words, although HIF-1α and CITED2 have identical affinities, in a competition CITED2 is able to completely displace HIF-1α.

The explanation involves an allosteric structural change in TAZ-1, linked to binding of both partners at multiple sites [33]: it clearly helps that both partners are flexible [35]. In the complex between TAZ-1 and HIF-1α, the HIF-1α wraps itself around TAZ-1, making several interactions, of which the strongest involve HIF-1α helices αB and αC (Figure 4). The linker between αB and αC is flexible [36]. CITED2 is able to bind to this complex to form a ternary complex by displacing the very weakly bound HIF-1α helix αA (Figure 4), but leaving HIF-1α interacting via αB and αC as well as the LPQL binding motif. This binding of CITED2 helix αA’ produces an allosteric change in helices α1 and α4 of TAZ-1, which enables CITED2 to bind more tightly and weakens the interaction with HIF-1α.

At this point comes the critical part of the facilitated dissociation process. If the LPQL binding motif of HIF-1α dissociates transiently, as it must do frequently, then the equivalent LPEL motif of CITED2 is able to replace it. Once there, it is able to strengthen the binding of CITED2 and weaken the binding of HIF-1α, allowing CITED2 to displace HIF-1α completely.

It is reasonable to ask how it is that CITED2 can displace HIF-1α, but HIF-1α cannot displace CITED2. The authors of [35] suggest that the difference is that there is greater thermodynamic coupling (stronger cooperativity) between different binding sites in CITED2 than there is in HIF-1α, mainly because of the more flexible linkers in HIF-1α, i.e., the binding of CITED2 at one site increases the affinity for binding at other sites, whereas this is much less true for HIF-1α.

This remarkable mechanism is significant, because it shows how flexible ligands are able to “worm their way in” and displace competing ligands, seemingly defying expectations from macroscopic affinities. Equally significant is that it provides the incoming ligand with a means to increase the off-rate of the outgoing ligand. The flexibility of the incoming ligand thus not only increases its own binding rate but also accelerates the departure of the outgoing ligand. Once again, protein flexibility (critically, including the flexibility of the receptor [36]) has facilitated faster on- and off-rates. An elegant extension of this model was shown by Singh et al. [32], who studied the process by which *O*-acetyl serine (OAS) is able to displace the inhibitor serine acetyl transferase (SAT) on the enzyme *O*-acetyl serine sulphhydrylase (OASS), even though SAT binds approximately 10^5^ times more strongly than OAS. Binding of OAS into the active site channel of OASS produces allosteric changes that reduce the affinity of OASS for SAT, and allow OAS to work its way into the active site via a series of conformational changes in the enzyme, for which they suggest the name “competitive allostery”.

### 2.6. Multiple Interactions Extend the Power of Avidity through Allovalency

The concept of allovalency was first proposed to explain the binding of the cell cycle protein Cdc4 to its intrinsically disordered inhibitor Sic1 [37]. This binding has a markedly non-linear dependence on the number of phosphorylations on Sic1: when Sic1 is phosphorylated by its kinase CDK on five or fewer sites, it binds weakly, but as soon as it is phosphorylated on at least six sites, its affinity increases dramatically. This phenomenon was explained (approximately) by a model in which all of the phosphates bind with equivalent affinities, and where phosphate-Cdc4 interactions exchange very rapidly, with no single interaction predominating (Figure 5). This gives rise to a very sharp (approximately sixth order) dependence on kinase concentration, and thus a sharply defined “on” switch as the kinase concentration increases. This concept, where the ligand interacts using multiple equivalent motifs and has no defined structure even when fully bound, was termed allovalency [37]. Equivalently, it has also been called ultra-sensitivity (see, for example, [38]). For intrinsically disordered ligands, one consequence is that the ligand is retained close to its receptor (in what was defined above as an encounter complex). Indeed, it can even dissociate completely, after which there is a high probability that one of the phosphate sites will rebind before the ligand has been able to diffuse away—an idea reminiscent of the fly-casting idea mentioned above. Many of the examples of this behaviour involve multiple phosphorylations, where as a consequence the ligand builds up a high charge density. In such cases, the retention of the ligand close to its receptor is helped both by electrostatic interactions (which act both to attract the ligand and receptor together and to prevent the ligand dissociating fully [11]) and by the high charge density on the ligand, leading to reduced conformational entropy [38]. We note that Ahmed et al. [31] observed facilitated dissociation/competitive allostery in this system, where phosphorylated peptides can bind to an allosteric site close to the phosphate binding site, which reduces the affinity at the binding site and produces more rapid dissociation (Figure 5). The overall binding affinity of phosphorylated Sic1 to CDK is increased by allovalency from around 100 μM for a single interaction to under 1 μM for the hexaphosphorylated interaction [38].

The last three sections can be summarised as saying that an interaction can affect the rate and strength of a neighbouring interaction, either positively or negatively, and this effect is relatively greater when each interaction is weak. Biological systems have used this simple idea to produce a wide range of outcomes, which are mostly aimed at ensuring rapid dissociation.

### 2.7. The Ultimate Complexity: Fuzzy Complexes

In this paper, we have been building from simpler to more complex types of interactions. The ultimate degree of complexity is provided when both receptor and ligand have multiple binding sites [39]. Typically, the ligand will be intrinsically disordered, as in allovalency, and have no fixed structure (although, as discussed above, the ligand is probably more structured in such complexes than it is when free [11]). This type of interaction has been described as fuzzy binding, by analogy to fuzzy logic, where individual interactions are best described not as “yes” or “no”, but using a probability. Individual interactions are always weak, but the overall strength of binding can be quite large because of the large number of interactions that cooperate multiplicatively, rather than by addition. Because each interaction is weak, its strength (and thus eventually the strength of the entire interaction) can be affected in a major way by the context, such as other neighbouring sequences, post-translational modifications, or other weak binding [11,40]. In an ensemble of molecules, each interaction is likely to have different local effectors, giving rise to a wide spread of “affinities”, making fuzzy complexes more like a rheostat or volume control than an on/off switch [39]. We note that the term “fuzzy binding” has been used to mean several different things: we prefer to restrict it to binding where both interacting partners can bind at several sites [39].

An instructive example of all these factors at work is provided by the interaction between nucleoporins (Nups) and nuclear transport receptors. The nuclear membrane contains multiple nuclear pores, which are approximately 60 nm in diameter. These pores need to be permeable to the many molecules that need to go into and out of the nucleus, but prevent free diffusion of proteins. They do this by having a high density of intrinsically disordered proteins called Nups. These proteins have multiple occurrences of the dipeptide phenylalanine-glycine, and are therefore often called FG-Nups. They form a dense “kelp bed” [41] that acts as a barrier to prevent free diffusion of proteins, and most of the proteins that pass through nuclear pores do so bound to nuclear transport receptors (importins and exportins, whose direction of travel is regulated by GTP). The transporters are specialised to be able to move rapidly through the pores, because they bind weakly to FG-Nups via the FG repeats, and thereby move “hand over hand” through the pores, being passed on from one FG repeat to the next. Each transporter has multiple sites that can bind FG repeats [42], implying that these interactions are truly fuzzy, although the details may differ for some Nups [43]. These interactions have many similarities to allovalency. Each individual interaction is very weak (measured as 7.3 mM for one interaction [44]), and has very fast on- and off-rates. The speed is critical: the distance across the barrier is approximately 30 nm, and yet the transit times are typically as short as 5 ms [44]. The total concentration of FG repeats within the pore is approximately 50 mM, giving rise to an overall affinity of transporter to FG-Nups in the nanomolar region [44]. Numerical estimates provided by [44] suggest that the individual off-rates are roughly five times faster than would be expected based on the affinity and diffusion-limited on-rate, which implies that there may be some facilitated dissociation occurring in order to speed up the transit.

## 3. Conclusions

We have emphasised that many transient protein–protein interactions are optimised for rapid on- and off-rates, rather than tight binding. We have not paid much attention to the specificity of the interaction. One would expect that specificity is inversely related in some way to the overall tightness of binding and also to the rate of binding and dissociation. The argument made here is that fast rates are more important than tight binding, which would imply that specificity of binding is not critical per se, and must be achieved in other ways. It is, for example, clear that in many signalling pathways, specificity is achieved by the assembly of multiple components into the same location, supported by multiple interactions [45]. This is essentially the function of the multiple adaptor and scaffold proteins that are often found in eukaryotic signalling pathways.

Many of the advances in understanding discussed here have been prompted by experimental investigations of IDPs. I suggest that the time is now ripe to turn to computer simulations to explore the effect of experimental variables over a wide range of values. Such simulations could to help to understand the importance of cellular crowding, cell infrastructure, very weak interactions, liquid-phase separation, and much more. Computer simulations can span a wide range of scales. Of particular interest is agent-based modelling [46], which can use modern high-speed GPUs to model interactions in parallel at high speed.

We have shown that rapid binding and dissociation are greatly helped by flexibility, both in the receptor and the ligand. In particular, protein ligands that contain intrinsically disordered regions provide multiple benefits, most importantly in ensuring fast on- and off-rates. The title of this paper is intended to highlight the importance of fuzzy complexes, but also to show that the binding of a protein to its ligand is determined by much more than merely its affinity, and that affinity is often not a very meaningful concept.

## Figures and Tables

**Figure 1 life-13-00855-f001:**
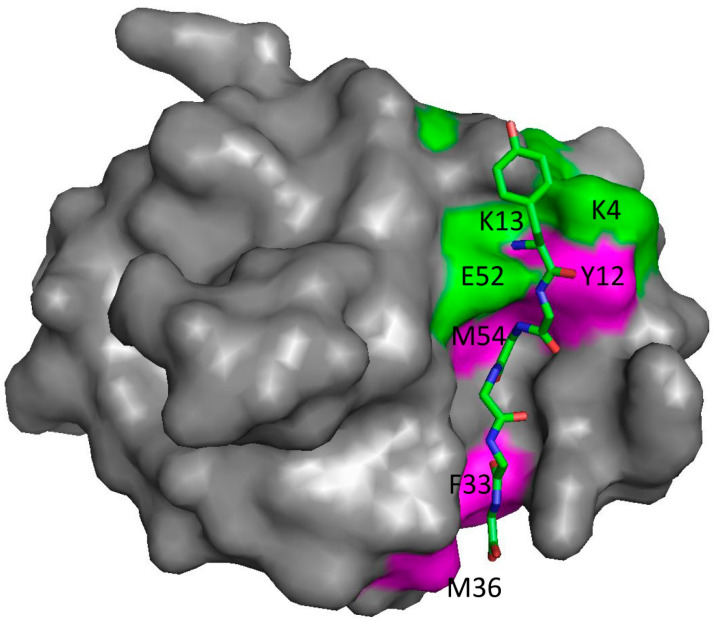
The structure of the protein SH3b bound to the peptide YGGGGG, based on Protein Data Bank coordinates 5leo [23]. YGGGGG is shown as sticks. The pentaglycine part is held rigidly in a groove on the protein surface, while the N-terminal tyrosine is not part of the biological ligand and is exposed on the surface and more dynamic. Residues marked in green are those measured by NMR to have an affinity significantly weaker than the average over the whole protein, while those in magenta have an affinity significantly stronger. Residues in grey have average affinities or could not be measured. Data from [22].

**Figure 2 life-13-00855-f002:**
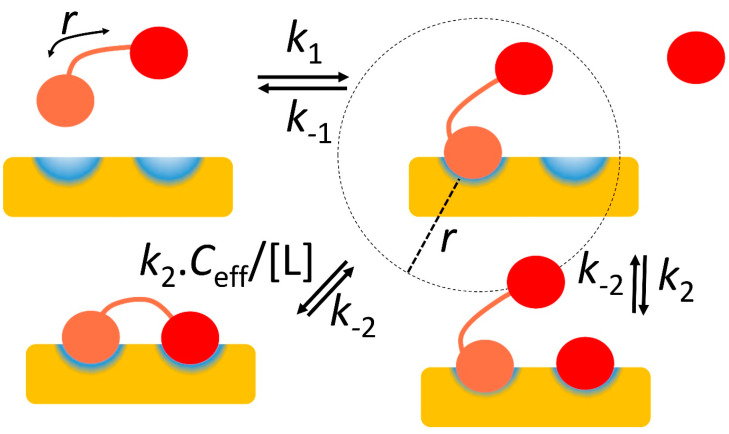
Avidity and effective concentration *C*_eff_. The protein receptor is in yellow, with two binding sites in blue. The ligand binds using the salmon and red domains, which in isolation have on-rates *k*_1_ and *k*_2_ respectively, and are connected by a linker of overall length *r*. After the ligand has bound using the salmon domain, the red domain may bind either using an unconnected binding domain, with on-rate *k*_2_, or the linked domain, with on-rate *k*_2_*C*_eff_/[L], where *C*_eff_ is the effective concentration of the red domain. This may be approximated by placing one molecule in a sphere of radius *r*, and is often very much larger than [L], thereby accelerating the on-rate and strengthening the binding considerably.

**Figure 3 life-13-00855-f003:**
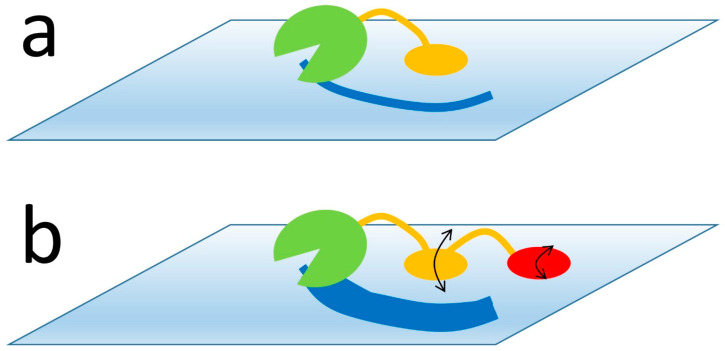
A typical polysaccharide-digesting enzyme has a catalytic domain (green) and one or more carbohydrate binding domains (orange/red), which attach the enzyme to the polysaccharide surface and prevent it from diffusing away before it has completed its digestion (dark-blue stripe). (**a**) With a single binding domain, when the enzyme has digested everything within reach, the binding domain must detach and the enzyme has to diffuse away and attach to a new site. (**b**) However, with two domains that individually bind more weakly but together bind with the same overall affinity, the two binding domains are able to “walk” across the polysaccharide surface without fully detaching (black arrows), and thus allow the enzyme to access many more substrate sites without having to dissociate.

**Figure 4 life-13-00855-f004:**
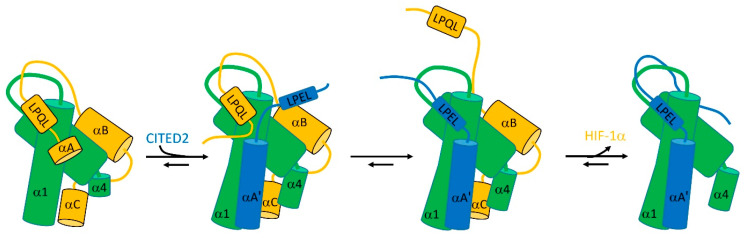
The facilitated dissociation that allows CITED2 (blue) to displace HIF-1α (orange) from TAZ-1 (green). See text for details. Figure adapted from [33].

**Figure 5 life-13-00855-f005:**
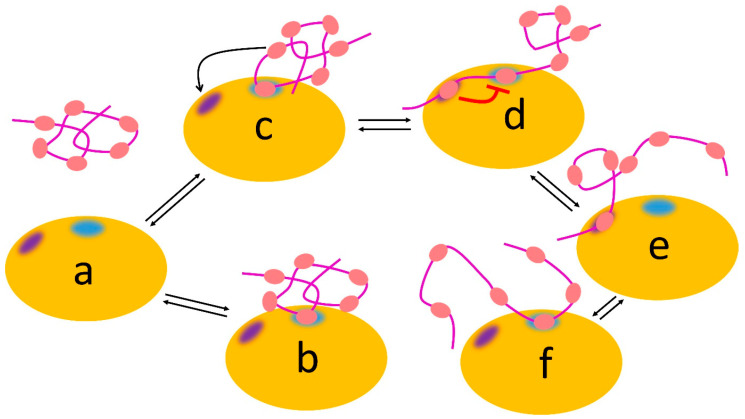
Allovalency. Schematic view of Cdc4 (orange), which has two sites: a binding site for phosphate (blue) and an inhibitory allosteric site (indigo). The Sic1 ligand (magenta) can bind using any one of its six phosphates (salmon), which bind weakly at the blue binding site, with no phosphate dominating the interaction (for example, complexes (**b**,**c**) equilibrating with free (**a**). If another phosphate binds at the allosteric site (**c**,**d**), it weakens binding of the phosphate at the blue site (red line), which leads to enhanced dissociation (**e**) followed by binding of another phosphate (**f**).

## Data Availability

Not applicable.

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
