# Peer review of "Protein Binding: A Fuzzy Concept"

_life, 2023, doi:10.3390/life13040855_

Round 1

Reviewer 1 Report

In this manuscript, Dr. Williamson discusses different aspects of protein binding. The main hypothesis is that the key driver of protein interactions is to maximize the dissociation rate. Various examples are provided that should support this hypothesis.

I am not convinced about the arguments provided. The author presents a narrow view of the field. Yes, dissociation rates can play a key role but I doubt that they are generally more important than affinity and specificity.

Here few points to consider and discuss.

1.     The author seems to focus on transient interactions of signaling proteins. However, there are also permanent or obligatory protein interactions (e.g. large complexes). Interactions in this case are often of high affinity. Maybe not as high as streptavidin/biotin but higher than in the cases discussed. Evolutionary constraints on this type of interactions are likely different than the ones acting on the proteins discussed.

2.     Even for signalling proteins, high affinity interactions can be important. Hormone receptors like the beta adrenergic receptor on the cell surface need to sense femtomolar concentration changes ( e.g. upon stress). This is only possible with high affinity binding.

3.     Point two highlights that the physiological concentration of the ligand/partner is very important. Receptors/ interaction partners ought to have affinities close to that concentration for binding to occur. Yes, if the affinity is in the range of the physiological concentration of the ligand, on/off rates can be tuned (by some of the mechanisms discussed), but stating that rates are generally more important than affinity seems incorrect.

4.     I agree that for many signaling interactions the tuning of kinetics is important, but I would avoid the generalization.

5.     Not all IDR-mediated interactions are of medium affinity. Some of these interactions are in the femtomolar range.

6.     In section 2.3, it is stated that “A more ordered bound state would produce an overall stronger affinity”. This is not generally true. Entropic aspects are not considered here. A major decrease in dynamics upon binding could reduce entropy and lower affinity.

7.     The author provides nice examples of systems with multiple, tethered binding modules/motifs. However, it is not discussed that such systems have also evolved to increase specificity. A good example is C-src kinase with a SH2 and SH3 domain, where binding of both domains increases target specificity. I don’t see how one can a priori exclude specificity as key player in the evolution of this protein.  

8.     In section 2.7, it is suggested that fuzzy interactions are “volume control” or rheostat-like and not switch-like. That is not correct as both types have been associated with fuzzy interactions. 

Author Response

Pleasse see the attachment, which is a response to both reviewers.

Reviewer 2 Report

Comments on “Protein binding: a fuzzy concept” by Mike P Williamson

In this manuscript, Williamson shows that on and off binding rates are better indicators of protein-substrate affinity than protein-substrate binding strength. I appreciate how the author provides a description of binding processes that directly addresses the dynamic nature of these systems. We often fall into a sterile view of life, where rigid puzzle pieces fit together in well-defined structures, but the chaotic, messy reality is an important feature of how life has evolved to meet the challenges of operation. Since protein systems are so floppy and dynamic, reductionist characterization of their behavior with just one or a few parameters is going to lose something. Therefore, perspectives on protein-substrate binding can provide complementary descriptions of these processes that allow easier highlighting of governing physics. I think this manuscript provides a useful and unique view of protein operation and should probably be published. I do have a few comments for the author to consider, which might enable improvement of the manuscript. 

  • I often think of the rate of a process being governed by the barrier that a process must overcome and the frequency of approaching the top of the barrier. It seems like such a description of these processes could connect this “fuzzy” view to a more traditional binding strength perspective.

  • This perspective seems similar to the framework of protein folding provided by Markov state models and understanding folding from many simulations of separate portions of the folding process. I wonder if connections could be made to the “stitching” together of many simulations into a concerted picture of the folding process in terms of the time scales connecting various states? It seems like this might provide a greater sense of perspective on how this manuscript fits in with larger questions of protein kinetics. 

  • Diffusion limited protein function is mentioned a few times, but it is not clearly defined. I am particularly interested 

  • I think a more direct definition of avidity at the beginning of section 2.4 would be helpful. Moving Fig. 2 up could provide a visual point for discussing this phenomena. 

  • I wonder if there are specific cases that could clarify the importance of quantifying binding strength vs. binding rate? In particular, this could connect well with molecular modeling of binding processes. I guess I am curious to see what the author thinks would be useful next steps for clarifying how binding occurs and what simplified metrics can be used to best understand a particular process. This might be useful at the end of the discussion section. 

Author Response

Please see the attachment, which is a response to both reviewers

Round 2

Reviewer 1 Report

The author addressed all my concerns